# Lymphatic embolization for early post-operative lymphatic leakage after radical cystectomy for bladder cancer

Yoo Sub Shin[1], Kichang Han[2], Jongsoo Lee[1], Hyun Ho Han[1], Won Sik Jang[1], Gyoung Min Kim[2]* , Ji Eun Heo[1]* 

1 Department of Urology, Urological Science Institute, Yonsei University College of Medicine, Seoul, Republic of Korea, 2 Department of Radiology, Research Institute of Radiological Science, Severance Hospital, Yonsei University College of Medicine, Seoul, Republic of Korea

☯ These authors contributed equally to this work.
* heoji87@yuhs.ac (JEH); gyoungmin@yuhs.ac (GMK)

## Abstract

**Data Availability Statement:** As this study covers an extremely specific population, our data set is potentially identifying patient information even after proper de-identification. Still, our data can be

### Background and objective

Although radical cystectomy (RC) with pelvic lymph node dissection (PLND) is the standard treatment of muscle invasive bladder cancer, it may cause lymphatic leakage. Recent studies describe lymphatic embolization (LE) as an option to manage post-operative lymphatic leakage. Hence, this study evaluated the outcome of LE in patients receiving RC and analyzed factors associated with outcomes.

### Methods

This was a retrospective analysis of patients who underwent LE after RC for bladder cancer between August 2017 and June 2023. The data was assessed for analysis at January 2024. The patients were divided into a clinical success group and a clinical failure group. Clinical failure was defined as the following: 1) those who required drainage catheter placement >7 days after LE, 2) those who needed re-intervention before catheter removal, and 3) those who experienced adverse events associated with LE. Logistic regression analysis was performed to identify the factors associated with outcomes of LE.

### Key findings and limitations

We analyzed 45 patients who underwent LE after RC. Twenty-eight (62.2%) patients were identified as clinically successful. Four patients required re-embolization, but none required more than two sessions of intervention. Three patients experienced lymphatic complications after LE. In multivariable analysis, maximal daily drainage volume of >1,000 mL/day (odds ratio [OR] = 4.729, 95% confidence interval [CI]: 1.018–21.974, p = 0.047) and diabetes mellitus (DM) (OR = 4.571, 95% CI: 1.128–18.510, p = 0.033) were factors associated with LE outcome.

shared upon request. Contact for our data can be made through the institutional review board (Yonsei-IRB, irb@yuhs.ac).

**Funding:** The author(s) received no specific funding for this work.

**Competing interests:** The authors have declared that no competing interests exist.

## Conclusions and clinical implications

Our results suggest LE as a potentially effective procedure for controlling post-operative lymphatic leaks after RC, with few minor side effects. Patients exceeding a daily drainage of 1,000mL/day or with a medical history of DM have a higher risk for re-intervention and clinical failure after LE.

## Introduction

Pelvic lymphatic leakage is a well-recognized postoperative complication after pelvic surgery. In most cases, these leaks resolve spontaneously because the rate of fluid leakage does not exceed the rate of absorption by the peritoneum. However, in cases of massive leakage, patients may require extended placement of post-operative drainage catheters, which can delay their overall recovery and initiation of adjuvant therapy. Additionally, excessive fluid build-up from these leaks can lead to the formation of lymphocele, resulting in secondary complications, e.g., infections, pain, and deep vein thrombosis [1].

For well-contained lymphatic fluid collections such as lymphoceles, percutaneous drainage and sclerotherapy using ethanol injection are effective. Injected ethanol has been proposed to cause fibrosis of the lymphatic channels, sealing leakage points [2, 3]. Early postoperative leakage, however, often occurs as a spill into the pelvic space, making it unsuitable for sclerotherapy [4]. Consequently, lymphatic embolization (LE) has garnered attention as an effective alternative to sclerotherapy. Not only is LE suitable for controlling early spillages, but it is also more effective than ethanol sclerotherapy in controlling leaks [5, 6].

Although LE has generally been considered safe and effective in multiple retrospective studies, no studies have focused on a single disease entity or type of surgery [4–7]. Pelvic surgery encompasses a wide range of urological, gynecological, and gastrointestinal surgeries, each with distinct patterns of nodal metastasis and surgical protocols.

Patients with muscle-invasive bladder cancer are frequently treated using radical cystectomy (RC) with bilateral pelvic lymph node dissection (PLND) [8]. Although the ideal extent of dissection has not been defined, the SWOG 1011 trial recently confirmed lower mortality and morbidity using a standard PLND template than that with an extended approach, without compromising overall patient survival [9]. Lymphatic complications may occur in up to 3–11% of patients, with an increased incidence in patients who undergo a wide PLND template or extraperitoneal approach [10–15]. However, the management of post-RC lymphatic leaks and/or lymphoceles is not well-defined. Only few case reports are available, each describing a different treatment approach [16, 17]. Such lymphatic complications can impede perioperative recovery and delay future treatment. Therefore, this study aimed to assess the efficacy of LE and identify the risk factors associated with successful outcomes of LE for lymphatic leakage after RC.

## Materials and methods

### Ethics statements

This study was approved by the Institutional Review Board (IRB) of Yonsei University Hospital (IRB number: 4-2023-0766). The requirement for informed consent was waived by the IRB owing to the non-invasive and retrospective study design. Data was provided to the authors after de-identifying all personal information of participants.

## Study design and population

We reviewed a single-institution database in January 2024 to identify patients who underwent LE after RC and PLND for bladder cancer between August 2017 and June 2023. All RC was performed with a curative intent. To ensure the inclusion of cases specific to early postoperative lymphatic leakage, the following criteria were applied: 1) lymphocele present prior to LE, 2) presence of idiopathic lymphatic leaks unrelated to PLND, and 3) postoperative leaks related to surgical procedures in anatomical regions outside the pelvis. To maintain the homogeneity of the cohort, we excluded patients with pathological reports describing conditions other than urothelial carcinoma. We defined the primary outcome of the study as the rate of clinical success of LE. Secondary outcomes included risk factors associated with LE and complications associated with the procedure.

Lymphatic leakage was confirmed when radiographic evidence of leakage was found in lymphangiography. Lymphangiography and LE was consulted in patients who continued to experience daily catheter drainage exceeding 500 mL/day for >5 days despite receiving conservative care. Conservative management included low-fat diet with medium-chain triglycerides (MCTs) and protein supplementation. Secondary causes of fluid collection such as infection, postoperative bleeding, and urine leakage were excluded prior to consultation. Consequently, we only included patients whose drainage fluid exhibited a grossly clear, odorless character, negative culture results, and creatinine levels consistent with their serum levels. Simultaneous LE following lymphangiography was performed when radiographic evidence of lymphatic leakage was found.

## Radical cystectomy and pelvic lymph node dissection procedure

Prior to surgery, patients received thorough consultation on the surgical approach and type of urinary diversion. Open RC was performed in an extraperitoneal fashion, and robotic RC was performed intraperitoneally using a multiport robotic system. All patients received a bilateral PLND using a standard template, dissecting the external iliac, internal iliac, and obturator lymph nodes. All lymph nodes were dissected proximal to the common iliac bifurcation, above the circumflex iliac vein and medial to the genitofemoral nerve. Bilateral iliac arteries, iliac veins and obturator nerves were skeletonized during PLND. Bipolar cauterization was used to seal visible leakage sites after PLND, with or without applying metal clips based on the surgeons' preference. A drainage catheter was placed in the pelvic space before closing the incision, and a post-operative abdominal X-ray was filmed, which confirmed its correct position.

## Lymphangiography, LE procedure

Two experienced interventional radiologists performed the LE procedure. Both inguinal lymph nodes (LNs) were punctured using a 3.5-cm long 25-gauge fine needle under ultrasound guidance. Lymphangiography was performed by manually injecting ethiodized oil (Lipiodol; Guerbet, LLC) into the LNs at an injection rate of 0.2–0.5 mL per minute until opacification of the pelvic lymphatic system was observed. Following the identification of lymphatic leakage, 5% dextrose water was injected into the LN before embolization to avoid premature polymerization. LE was performed by injecting a mixture of N-butyl cyanoacrylate and ethiodized oil at a ratio of 1:1–1:4 into the LNs, which provided afferent lymphatic flow to the leakage sites (Fig 1). When there was a LN closer to the leakage site than the initially accessed LN, that closest upstream LN was additionally punctured and embolization was performed there.

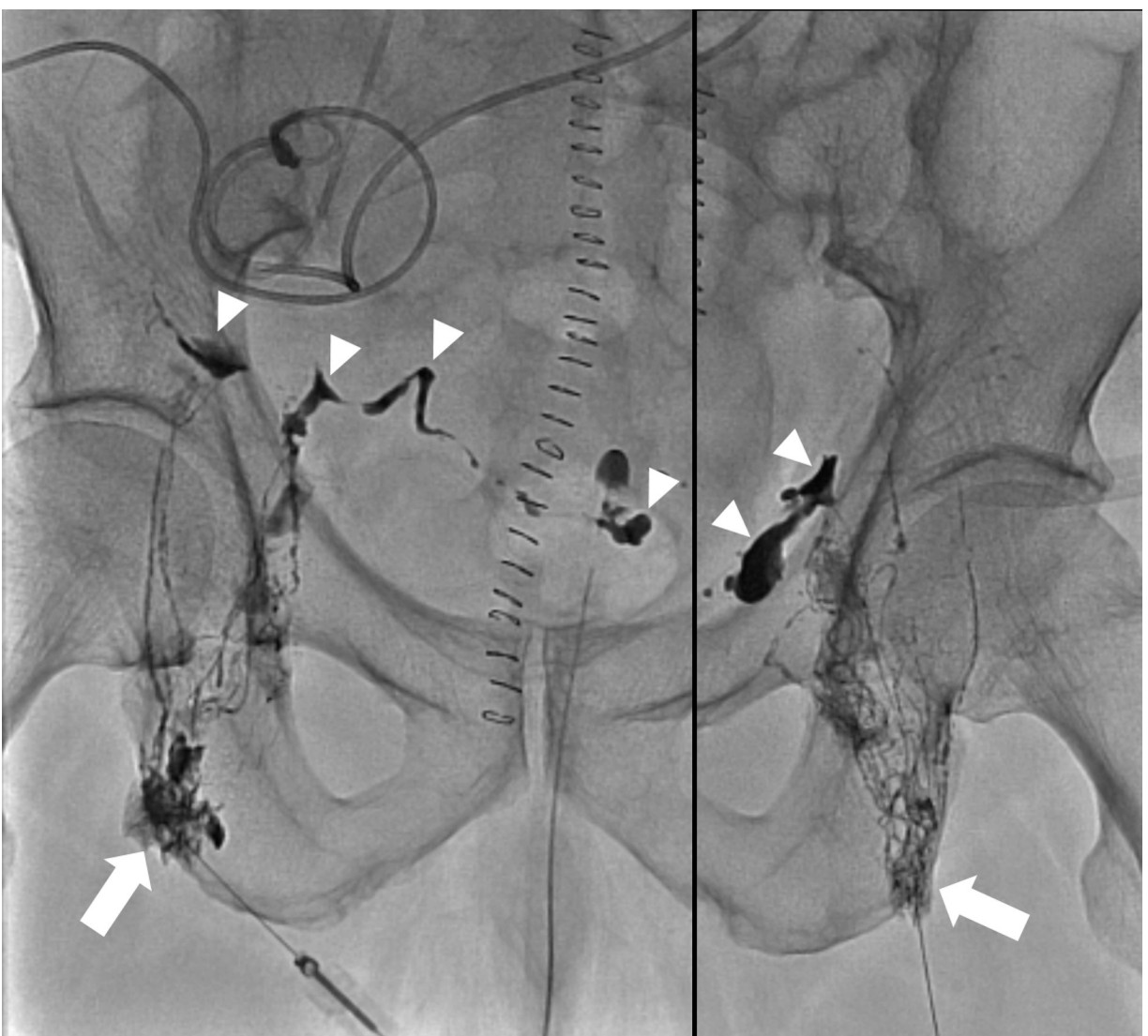

**Fig 1. Successful clinical outcome in a 67-year-old man.** Daily catheter drainage was increasing up to 1600 mL/day. Lymphatic embolization was performed on postoperative day 5. Leakage of Lipiodol into the pelvic cavity was noted (arrowheads) bilaterally on nodal lymphangiography. Embolization was performed by injecting a mixture of N-butyl cyanoacrylate and Lipiodol into the lymph nodes (arrows) at a ratio of 1:3. The amount of drainage decreased to 32 mL/day at 2 days after procedure, and the drainage catheter was removed.

### Data collection

We collected the following data: the daily drain output before and after LE, patient age at the time of the procedure, body mass index, sex, medical history, surgical approach, type of urinary diversion, estimated blood loss (EBL), and pathological results. Lymph node density was calculated as the percentage of positive lymph nodes from the total number of nodes dissected during PLND ([number of positive nodes/number of total dissected nodes] *100).

All patients received imaging studies and out-patient follow-ups based on the National comprehensive cancer network bladder cancer guideline (version 4.2024) [18]. Image studies and out-patient records within 6 months after LE were reviewed for LE associated complications.

After LE, patients were categorized into clinical success and failure groups. Although the definition of clinical success varied upon different studies, we applied a unique and stringent

for defining clinical success [5, 6]. Patients were included in the failure group if they met any of the following criteria: 1) drainage catheter placement required > 7 days post-LE, 2) re-intervention required prior to catheter removal, and 3) occurrence of adverse events associated with LE, e.g., lower extremity edema found on computed tomography (CT) within 6 months of post-LE follow-up. There was no consistent timing for drainage catheter removal among the patients, as the timing was determined by the surgeon in charge of the patient; however, drainage removal was frequently indicated when daily drainage was < 300 mL per day. Re-intervention was defined as the requirement for more than one session of LE prior to drainage catheter removal.

## Statistical analysis

Comparative analysis was conducted using the Fisher exact test; linear-by-linear association analysis was used to analyze categorical variables, and the Mann–Whitney test was used to analyze continuous variables. Additionally, logistic regression analysis was performed to identify the risk factors for clinical failure. The level of significance was set at $p<0.05$. The cutoff value for drainage volume used in logistic regression was set in reference to previous studies [19, 20]. All statistical analyses were performed using SPSS (version 26.0; IBM Corp., Armonk, NY, USA).

## Results

A total of 491 patients underwent RC during the study period, of whom 55 subsequently underwent LE. Ten patients were excluded from this study for the following reasons: non-urothelial cancer origin (n = 6), LE after drainage catheter removal (n = 2), non-procedure-related mortality (n = 1), or incomplete clinical data (n = 1) (Fig 2). In total, 45 patients were included in our analysis.

Table 1 shows the clinical characteristics of patients in the success and failure groups. Among the 45 patients included in this study, 28 (62.2%) were categorized into the success group. Diabetes Mellitus (DM) was more prevalent in the clinical failure group (21.4% and 52.9%, p = 0.050). Other demographic variables were not significantly different between the two groups. Moreover, the surgical approach (p = 0.144), urinary diversion type (p = 0.341), and EBL (p = 0.927) did not differ between the groups. The median number of dissected LNs were 16.5 (12.3–24.5) and 19.0 (14.5–25.5) in the success and failure groups, respectively (p = 0.331). with both groups representing similar frequencies of nodal metastasis (42.9% and 41.2%, respectively; p = 1.000). Distant metastasis was found only in the success group (17.9% vs. 0%, p = 0.140).

The average daily drainage volumes were 487.8 mL/day and 613.0 mL/day in the success and failure groups, respectively (p = 0.058). The maximum daily drainage was significantly higher in the failure group than in the success group (750.0 mL/day vs. 1014.0 mL/day, p = 0.050). Fig 3 depicts the median drainage volume of the clinical success and failure group, from 5 days prior to LE to 5 days post-LE. An overall decrease in volume after LE is observed in both groups. 24 of 28 (85.7%) clinical success group removed their drain within 5 days after LE, and 4 out of 17 (23.5%) clinical failure group removed their drain within 5 days after LE.

Among the 17 (37.8%) patients classified in the failure group, 4 required re-embolization due to recurrent lymphatic leakage, and 10 either failed to have their drainage catheter removed prior to discharge or received delayed removal of their drains (Table 2). Of the 4 patients requiring re-embolization, all successfully removed their catheter after their second session. Three patients presented lower extremity edema after LE. All edemas were self-resolved during out-patient follow-up. Five patients were found with incidental, asymptomatic

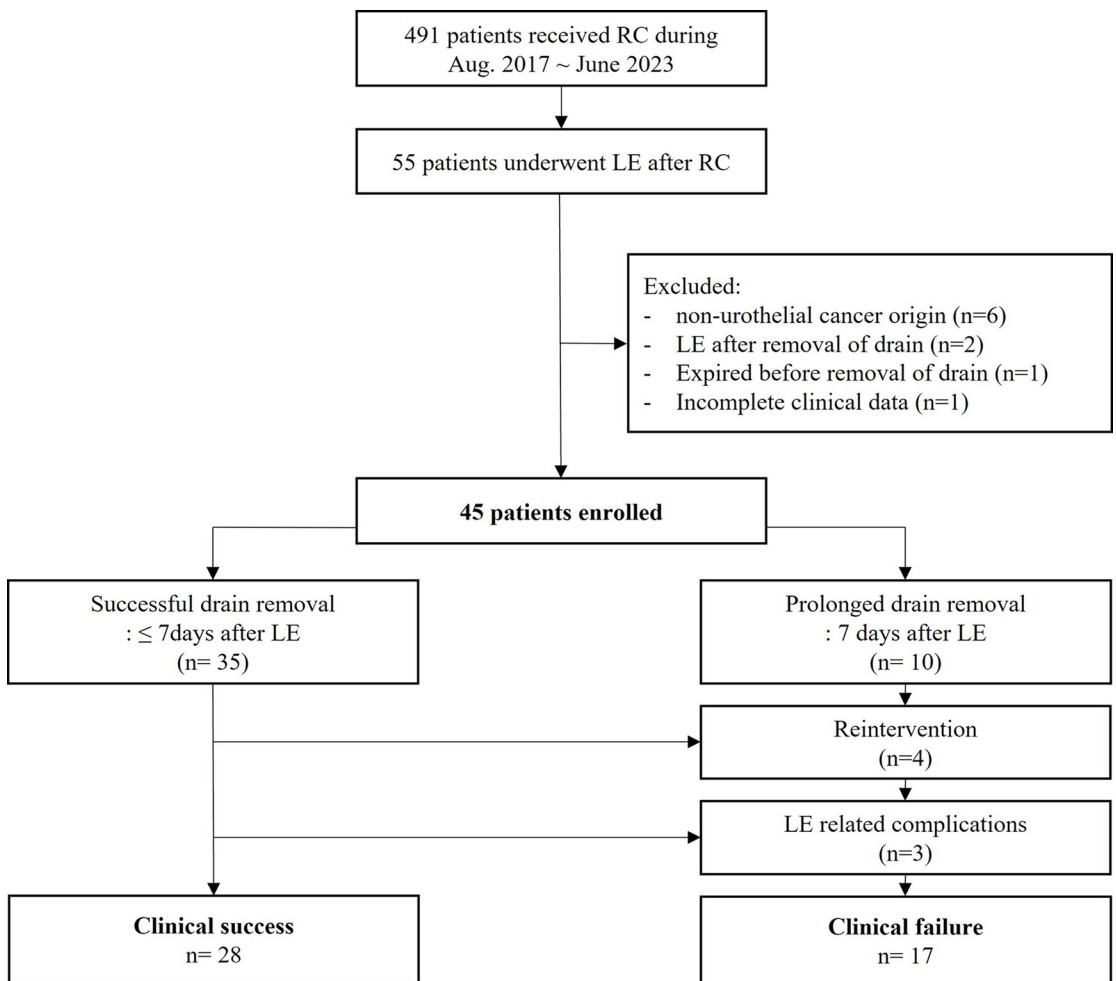

**Fig 2. Summary of patient selection and results of lymphatic embolization for lymphatic leakages.**

lymphoceles after LE through routine follow-up abdominal CT. None required additional drainage. Lymphoceles were located either in the lower quadrant of the abdomen or the pelvic space, and the median size of the lymphocele was 4.0 cm. Although the largest lymphocele reached up to 10.0 cm, no complications or symptoms were found.

The results of the logistic regression analysis of the clinical outcomes of LE are presented in Table 3. Multivariate analysis demonstrated that maximal daily drainage volume of >1,000 mL/day (odds ratio [OR] = 4.729, 95% confidence interval [CI]: 1.018–21.974, p = 0.047) and diabetes mellitus (DM) (OR = 4.571, 95% CI: 1.128–18.510, p = 0.033) were significantly associated with clinical failure of LE. Nodal stage of disease (OR = 0.933, 95% CI: 0.275–3.168, p = 0.912) nor lymph node density (cut-off$\geq$20%) (OR = 0.923, 95% CI: 0.225–3.780, p = 0.911) was not associated with clinical outcome.

## Discussion

While previous investigations of LE have included heterogeneous patient groups with pelvic lymphatic leakage, our study marks the first attempt to explore LE exclusively in patients with bladder cancer undergoing RC. Of the 45 cases, 35(77.8%) patients underwent successful drainage removal within a week after LE, and all but 1 patient had their drain removed before

**Table 1. Patients' characteristics.**

| Characteristics | Success (n = 28) | Failure[a] (n = 17) | P value |
|---|---|---|---|
| Age (y) | 73.0 (69.3–77.0) | 74.0 (68.0–80.0) | 0.972 |
| Body mass index (kg/m$^2$) | 24.5 (22.7–25.5) | 23.4 (21.1–25.8) | 0.223 |
| Sex | | | 0.547 |
| Male | 27 (96.4%) | 15 (88.2%) | |
| Female | 1 (3.6%) | 2 (11.8%) | |
| Medical history | | | |
| Hypertension | 15 (53.6%) | 8 (47.1%) | 0.763 |
| Diabetes mellitus | 6 (21.4%) | 9 (52.9%) | 0.050 |
| ASA | | | 0.381 |
| 1 | 1 (3.6%) | 0 (0%) | |
| 2 | 6 (21.4%) | 7 (41.2%) | |
| 3 | 21 (75%) | 10 (58.8%) | |
| Surgical approach | | | 0.144 |
| Open | 27 (96.4%) | 14 (82.4%) | |
| Robotic | 1 (3.6%) | 3 (17.6%) | |
| Urinary diversion | | | 0.341 |
| Ureterocutaneostomy | 9 (32.1%) | 5 (29.4%) | |
| Ileal conduit | 18 (64.3%) | 9 (52.9%) | |
| Neobladder | 1 (3.6%) | 3 (17.6%) | |
| Estimated blood loss (mL) | 600.0 (462.5–1000.0) | 600.0 (300.0–1100.0) | 0.927 |
| T stage | | | 0.537 |
| ≤T2 | 10 (35.7%) | 8 (47.1%) | |
| >T2 | 18 (64.3%) | 9 (52.9%) | |
| N stage | | | 1.000 |
| N0 | 16 (57.1%) | 10 (58.8%) | |
| ≥N1 | 12 (42.9%) | 7 (41.2%) | |
| M stage | | | 0.140 |
| M0 | 23 (82.1%) | 17 (100%) | |
| M1 | 5 (17.9%) | 0 (0%) | |
| Variant histology | 11 (64.7%) | 6 (35.3%) | 1.000 |
| Positive surgical margin | 5 (17.9%) | 2 (11.8%) | 0.693 |
| Total number of dissected LN (n) | 16.5 (12.3–24.5) | 19.0 (14.5–25.5) | 0.331 |
| LN density (%) | 0.0 (0.0–18.3) | 12.0 (9.0–26.0) | 0.979 |
| Average catheter drainage (mL/d) | 487.8 (380.6–601.4) | 613.0 (474.5–833.0) | 0.058 |
| Maximum catheter drainage (mL/d) | 750.0 (583.5–1039.8) | 1014.0 (693.5–1525.0) | 0.050 |
| Duration from operation to LE (d) | 7.0 (6.0–9.75) | 7.0 (7.0–11.0) | 0.271 |
| Day from LE to drain removal (d) | 4.0 (2.0–5.0) | 8.0 (5.5–12.0) | <0.001 |
| Day to discharge after LE (d) | 4.0 (3.0–7.8) | 12.0 (9.0–26.0) | <0.001 |

ASA = American society of anesthesiologists physical status classification, LE = Lymphatic embolization; LN = lymph node. Data are expressed median (interquartile range) or N (%).

a) Failure defined as: 1) Requiring reintervention, 2) > 1 week to drainage catheter removal post lymphatic embolization, 3) lymphatic complications.

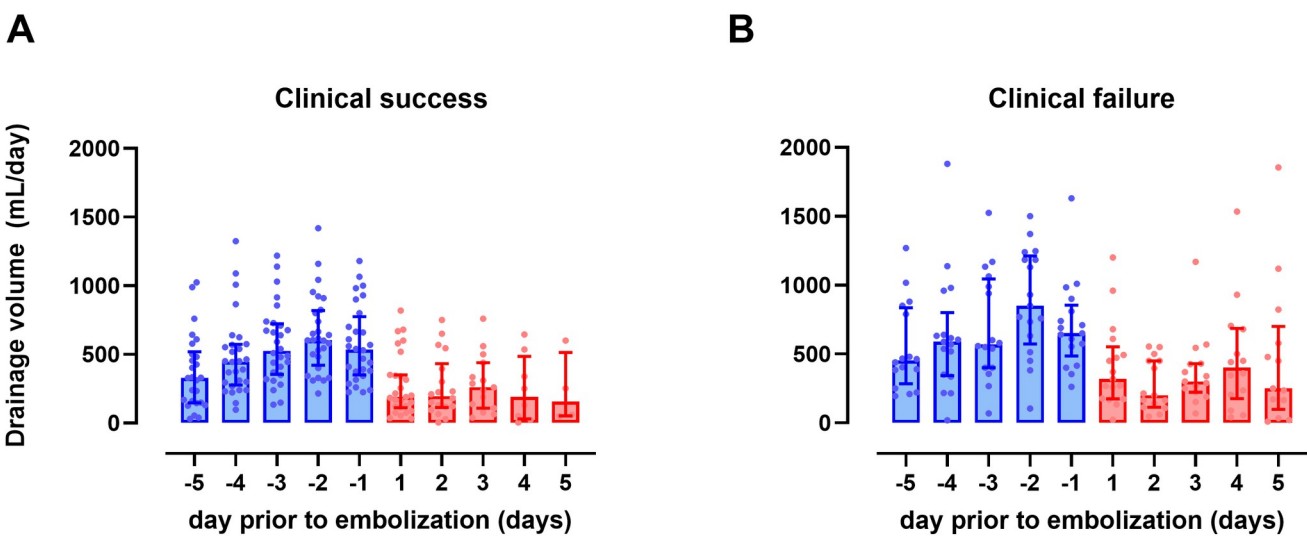

**Fig 3. Median daily drainage volume of patients from 5 days prior to LE to 5 days after LE.** A) Median daily drainage volume in clinical success group. B) Median daily drainage volume in clinical failure group.

discharge. While minor side effect (leg edema) was observed, no major adverse events were observed during the 6-month follow-up. Thus, our study demonstrates that LE is an effective and safe procedure for refractory lymphatic leakage in patients after RC.

Conservative management has traditionally been the primary approach for managing lymphatic leakage. Leibovitch et al. proposed an algorithm starting with MCT diets, followed by

**Table 2. Clinical outcomes of lymphatic embolization.**

| Outcomes | No. of events (%) |
|---|---|
| Reintervention | 4 (8.9%) |
| Catheter Removal | |
| Success ($\leq$ 1 week) | 35 (77.8%) |
| Delayed (> 1 week) | 9 (20.0%) |
| Failure | 1 (2.2%) |
| Intervention related complication | |
| None | 42 (93.3%) |
| Lower extremity edema | 3 (6.7%) |
| Non-lymphatic complication[a] | |
| Gastrointestinal | 4 (8.9%) |
| Pulmonary | 2 (4.4%) |
| Wound complication | 2 (4.4%) |
| Anastomosis stricture | 2 (4.4%) |
| Anastomosis leakage | 1 (2.2%) |
| Clinical success | |
| Success | 28 (62.2%) |
| Failure | 17 (37.8%) |
| Follow up imaging | |
| Asymptomatic lymphocele | 5 (8.9%) |
| Complicated lymphocele | 0 (0%) |

a) complications greater Clavien-dindo classification grade 3 were recorded

**Table 3. Logistic regression analysis of risk factors associated with clinical outcome.**

| 1. Univariate analysis | | |
|---|---|---|
| **Variables** | **OR (95% CI)** | ***P* value** |
| Sex | 3.60 (0.30–43.08) | 0.312 |
| Age (≥75 years) | 0.56 (0.10–3.16) | 0.511 |
| Body mass index (kg/m$^2$) | 2.67 (0.74–9.60) | 0.133 |
| Medical history | | |
| Hypertension | 0.77 (0.23–2.58) | 0.672 |
| Diabetes Mellitus | 4.13 (1.11–15.32) | 0.034 |
| Average catheter drainage (≥500mL/d) | 3.25 (0.85–12.45) | 0.085 |
| Maximum catheter drainage (≥1000mL/d) | 4.20 (1.00–17.60) | 0.050 |
| Surgical approach | 5.40 (0.44–66.67) | 0.188 |
| T stage (≥T2) | 0.63 (0.18–2.13) | 0.453 |
| N stage (≥N1) | 0.93 (0.28–3.17) | 0.912 |
| Variant histology | 0.84 (0.24–2.95) | 0.789 |
| Positive surgical margin | 0.61 (0.16–3.58) | 0.587 |
| Lymph node density (≥20%) | 0.92 (0.23–3.78) | 0.911 |
| 2. Multivariate analysis | | |
| Variables | OR (95% CI) | *P* value |
| Diabetes Mellitus | 4.57 (1.13–18.51) | 0.033 |
| Maximum catheter drainage (≥1000mL/d) | 4.73 (1.02–21.97) | 0.047 |

OR = odds ratio, CI = Confidence interval

the addition of total parental nutrition or somatostatin analogs [21]. In parallel with the afore-mentioned study, early investigations proceeded to surgical intervention only after the failure of dietary and medical management [22, 23]. However, conservative management often requires prolonged drainage catheter placement, which not only affects the patient's quality of life and prolongs hospitalization, but also increases the risk of secondary infection. Therefore, the advent of LE has enabled early intervention for refractory leaks, resulting in quicker resolution of lymphatic leakage and thus improving patient quality of life and complications associated with the leak.

Early studies have shown the efficacy and safety of LE for postoperative lymphatic leakage [1, 24–26]. However, initial impact was limited as early studies focused on the feasibility and technical aspect of the procedure. Recent investigations have attempted to compare LE with sclerotherapy; Kim et al. showed not only improved clinical success rates in LE compared with sclerotherapy, but also a reduced number of sessions in patients receiving LE [5]. A comparison by Seyferth et al. showed similar results, with earlier resolution of leakage in the LE group than in the sclerotherapy group [6]. Further, Lee et al. analyzed the risk factors for successful LE in 71 patients [4]. Their research revealed that old age and a larger drainage volume before LE (>1,500 mL/day) were associated with clinical failure.

In this study, we found that a maximum preprocedural drainage volume >1,000 mL/day was an independent predictor of LE failure. The average drainage volume showed a similar trend, although this did not reach statistical significance. Our results are consistent with those of a previous study that demonstrated that daily drainage of >500 mL/day negatively affects the clinical success of therapeutic lymphangiography [27]. Although the exact causality linking drainage volume to embolization remains uncertain, it is conceivable that a large drainage volume correlates with multiple leakage sites or severe lymphatic injury.

The absence of universal guidelines for managing lymphatic leakage underscores the clinical challenges that practitioners face. Balancing decisions between early intervention and conservative management remains pivotal for the treatment of patients with lymphatic leakage. Some studies have advocated an early surgical approach when preoperative leakage exceeds 1,000–1,500 mL/day, as larger drainage volumes are more resistant to conservative treatment than smaller drainage volumes [28–30]. Our study extended this paradigm to LE, suggesting the potential for early intervention in patients with large drainage volumes. Still, high risk patients with a high drainage volume or underlying DM should be informed of the increased risk of clinical failure.

Our study reported a lower success rate than that reported in previous research on LE (62.2% vs. 80–100%) [4, 6, 7, 26]. We attribute this difference to our re-defined criteria for clinical success. Prior studies on LE have defined clinical success on the basis of drain removal without recurrence. However, this definition overstates the importance of LE, given that lymphatic leakage can be self-limiting and may not always require intervention [19]. Consequently, we adopted a more comprehensive criterion by including only patients who had a procedure-to-drainage removal interval within 7 days, while excluding those requiring re-intervention or experiencing adverse events. When the definition of clinical success described in previous studies was applied, only one of the 45 patients failed to have their drainage catheter removed prior to discharge, yielding a success rate of 97.8% [4, 6, 7, 26].

DM is a well-known risk factor for lymphatic vascular integrity, as it disrupts the lymphatic endothelium and inhibits lymphangiogenesis [31–33]. Therefore, it was not surprising that DM also negatively influenced the success of LE in our study. Patients with bladder cancer are already predisposed to poor lymphatic vessel integrity, as they are predominantly diagnosed at an older age than are patients with other cancers. Patients are most commonly diagnosed in their 70s, with >55% of initial diagnoses occurring after 70 years of age [34, 35]. Age-related changes in lymphatic vessels include the degeneration of vascular walls and decreased contractility, which may contribute to increased permeability to lymphatic fluid [36, 37]. DM, therefore, may have worsened the already compromised lymphatic vessels, acting as a confounding risk for the success of LE.

This study has some limitations. The retrospective nature and small sample size introduced inherent bias. Moreover, as there is currently no consensus on the timing of intervention and drainage catheter removal, these decisions are made individually by the surgeon or hospitalist in charge of the patient's postoperative care. Nevertheless, our study is unique in that it is the first to investigate the efficacy of LE for a single disease. Therefore, we considered specific pathological factors and surgical methodologies in our analyses, which were not possible to consider in a heterogeneous patient group. Our study could further illustrate the characteristics of lymphatic leakage in bladder cancer, which has been poorly characterized in the past. Future prospective studies with larger cohorts and longer follow-up periods are warranted to define clear indications for LE and drainage catheter removal.

## Conclusions

Our results demonstrate LE as a potentially safe and effective intervention for refractory lymphatic leakage after RC. However, patients with underlying DM and patients with a daily drainage volume of >1,000 mL/day have a higher risk of clinical failure of LE.

## Author Contributions

**Conceptualization:** Yoo Sub Shin, Won Sik Jang, Ji Eun Heo.

**Data curation:** Yoo Sub Shin.

**Formal analysis:** Yoo Sub Shin.

**Investigation:** Yoo Sub Shin, Kichang Han.

**Methodology:** Hyun Ho Han, Won Sik Jang.

**Project administration:** Gyoung Min Kim, Ji Eun Heo.

**Validation:** Kichang Han, Jongsoo Lee, Hyun Ho Han.

**Writing – original draft:** Yoo Sub Shin, Jongsoo Lee, Ji Eun Heo.

**Writing – review & editing:** Yoo Sub Shin, Gyoung Min Kim, Ji Eun Heo.

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
