## [Decision Letter · Decision Letter 0]

11 Jul 2024

PONE-D-24-20978Lymphatic embolization for early post-operative lymphatic leakage after radical cystectomy for bladder cancerPLOS ONE

Dear Dr. Heo,

Thank you for submitting your manuscript to PLOS ONE. After careful consideration, we feel that it has merit but does not fully meet PLOS ONE’s publication criteria as it currently stands. Therefore, we invite you to submit a revised version of the manuscript that addresses the points raised during the review process.

**I suggest to consider the comments of the reviewers to improve the quality of the manuscript with the main focus on methodological details, sub-group analysis and statistics. **

We look forward to receiving your revised manuscript.

Kind regards,

Gernot Ortner

Academic Editor

PLOS ONE

Journal Requirements:

Reviewers' comments:

Reviewer's Responses to Questions

**Comments to the Author**

1. Is the manuscript technically sound, and do the data support the conclusions?

Reviewer #1: Yes

Reviewer #2: Yes

Reviewer #3: Partly

2. Has the statistical analysis been performed appropriately and rigorously? 

Reviewer #1: Yes

Reviewer #2: Yes

Reviewer #3: No

3. Have the authors made all data underlying the findings in their manuscript fully available?

Reviewer #1: Yes

Reviewer #2: No

Reviewer #3: Yes

4. Is the manuscript presented in an intelligible fashion and written in standard English?

Reviewer #1: Yes

Reviewer #2: Yes

Reviewer #3: Yes

5. Review Comments to the Author

**Reviewer #1:** This is a retrospective study of 45 patients who underwent LE following RCx for BCa. The success rate was 62.2%, and lymphatic complications after LE were noted in three patients. In MVA, maximal daily drainage of >1,000 mL/day and DM were independent factors associated with LE failure.

The paper is well-written with an appropriate methodology and a clear message.

Comments and questions:

Introduction:

1. When discussing LND extent in RCx, it would be beneficial to mention the results of the SWOG 1011 trial.

2. I recommend reporting further details on the management of lymphatic leakage in “bladder cancer” patients instead of presenting general data.

Methods:

1. Were all cystectomies performed with curative intent, or were there any palliative cases?

2. Please add details about LND. What methods were used for ligating the lymphatics (metal clips, bipolar, etc.)? How many drains in which locations were placed after the surgery?

3. Daily drainage < 300 mL/day has been the indication for drain removal. Please comment on this, as it is on the high side.

4. The primary and secondary outcomes should be clearly defined.

Results:

1. Given the number of cases, the continuous variables should be reported as median (IQR or range) instead of mean (SD).

2. Is there any data on baseline comorbidities (e.g., ASA, CCI, ECOG), length of hospital stay, and overall perioperative complications?

3. Among those who responded well to LE, what was the trend of lymphatic drainage before and after embolization?

Discussion:

1. The data reported on line 212 (77.6% successful drain removal) is a bit confusing.

2. Please highlight the pros and cons of LE over conservative management.

3. For those who are at higher risk for LE failure, what is your suggested strategy?

**Reviewer #2:** Dear Authors,

Thank you for submitting your manuscript. This study presents an interesting analysis of 45 patients who underwent lymphatic embolization after radical cystectomy. I appreciate the opportunity to review it. Below are my comments and suggestions for improvement:

- I would appreciate a more detailed description of the embolization procedure, providing more technical information.

- You only described perioperative variables of the cystectomy. Consider adding other variables, such as the quantity of embolizing fluid injected, etc.

- For the multivariate analysis, why did you set 1000 mL as a cutoff? Is there a study that defines that value as a predictor of lymphatic leakage? You should consider using a discrete value, for example, every 100 mL/day, to understand how the probability of the event increases with each increment.ù

Thank you again for your work and for allowing me to review your manuscript.

**Reviewer #3:** Shin and colleagues present a retrospective study on the outcome of lymphatic embolization (LE) in patients undergoing RC with PLND (standard PLND template) proximal to the common iliac bifurcation, inferior to the circumflex iliac vein and medial to the genitofemoral nerve. LE was performed if the daily catheter drainage exceeded 500 mL/day for >5 days despite conservative therapy. The authors excluded patients with lymphocele evident on CT before LE.

Patients were stratified by success vs. failure. Failure was defined as: 1) those who required drainage catheter placement >7 days after LE, 2) those who needed re-intervention before catheter removal, and 3) those who experienced adverse events associated with LE. LE was performed by injecting a mixture of N-butyl cyanoacrylate and ethiodized oil at a ratio of 1:1–1:4 into the inguinal LN.

Of 491 patients, 55 patients experienced a lymphocele with LE. 45 patients were included in this study. Of these, 28/45 were in the success group. The mean number of resected LN was 18.89 ± 9.53 and 20.94 ± 10.50 for the success and the failure group, respectively. The mean time from RC to LE was 8.50 ± 3.91 and 13.18 ± 17.22 days for the success and the failure group, respectively. Drainage catheters remained in place significantly longer in the failure group (3.54 ± 1.69 vs. 9.12 ± 5.12).

Secondary interventions in the failure group (n=17) comprised the need for a reintervention/-embolization, prolonged catheter, complications etc.

In the multivariable analysis, maximal daily drainage volume of >1,000 mL/d and diabetes were associated with clinical failure. The scientific English is good.

I have a few points to consider:

• The authors should describe the PLND template in more detail.

• What is the proportion of patients with lymphoceles that underwent LE?

• The authors excluded patients with lymphocele evident on CT before LE. What was the rationale for this?

• The authors state that (page 6, line 106): “Lymphatic leakage was diagnosed when radiographic evidence of leakage was found in lymphangiography“. Did all patients undergo lymphangiography?

• The authors excluded patients with evident lymphoceles on CT before LE. Why were those patients excluded? Why were not all patients included that underwent LE?

• The size and extent of the lymphoceles are poorly characterized. Please elaborate.

• How was correct catheter placement during drainage of the lymphocele verified?

• I suggest revising the statistics of the paper, adhering to the statistical outcome reporting guidelines provided by major urological journals

• How were the complications assessed? Was there a standardized approach?

• The confidence intervals in the multivariable analysis are incredibly large.

6. PLOS authors have the option to publish the peer review history of their article (what does this mean?). If published, this will include your full peer review and any attached files.

Reviewer #1: No

Reviewer #2: **Yes.**

Reviewer #3: No

---

## [Author Response · Author response to Decision Letter 0]

12 Aug 2024

Reviewer #1: 

We would like to thank you for your thorough evaluation of our paper. We revised our paper after reviewing your recommendations for revision: 

1. Introduction: When discussing LND extent in RCx, it would be beneficial to mention the results of the SWOG 1011 trial.

We added the SWOG 1011 trial information in our introduction, line 81-84. 

2. Introduction: I recommend reporting further details on the management of lymphatic leakage in “bladder cancer” patients instead of presenting general data.

We definitely agree with your comment. Despite lymphatic complications frequently occurring up to ~10% in RC patients, not much research has been conducted on the disease course of lymphatic leakages specific to RC. We therefore believe that our study is one of the first to identify lymphatic leaks specifically related to bladder cancer. We elaborated the management of leakage specific to bladder cancer in line 84-87.

3. Methods: Were all cystectomies performed with curative intent, or were there any palliative cases?

Yes, all radical cystectomies were performed with curative intent without any palliative cases. We have specified it in line 101-102.

4. Methods: Please add details about LND. What methods were used for ligating the lymphatics (metal clips, bipolar, etc.)? How many drains in which locations were placed after the surgery?

Thank you for the suggestion. We included the information regarding the methods used to seal the lymphatics in line 128-130.

5. Methods: Daily drainage < 300 mL/day has been the indication for drain removal. Please comment on this, as it is on the high side.

As mentioned in line 168-171, the timing of catheter removal was dependent on the surgeon in charge of the patient. 300mL/day was the common indication for drainage catheter removal in our hospital. We have rephrased the paragraph for clarity. 

6. Methods: The primary and secondary outcomes should be clearly defined.

We clarified our primary and secondary outcomes in 108-110. Thank you for the suggestion.

7. Results: Given the number of cases, the continuous variables should be reported as median (IQR or range) instead of mean (SD).

The following changes have been made in Table 1 and results. 

8. Results: Is there any data on baseline comorbidities (e.g., ASA, CCI, ECOG), length of hospital stay, and overall perioperative complications?

Thank you for the suggestion. We added details of ASA in Table 1. CCI and ECOG were not recorded during hospital stay. We also added the frequency of non-lymphatic peri-operative complications in Table 2. Clavien-dindo grade 3 or higher was counted in the analysis, as grade 1 and 2 complications were extremely frequent after RC. 

9. Results: Among those who responded well to LE, what was the trend of lymphatic drainage before and after embolization?

We have added Figure 3 to better illustrate this trend. Both the clinical success and clinical failure group showed a visible decline in drainage volume after LE. However, pre-procedural drainage volume was higher in the clinical failure group, and the clinical success group had more patients who removed their drains, while only a few patients in the clinical success group removed their drain within 5 days after LE (85.7% vs 23.5%). We have added this detail in Figure 3 and line 210-214.

10. Discussion: The data reported on line 212 (77.6% successful drain removal) is a bit confusing.

Thank you for pointing this out. We rephrased the statement (line 246-248), to clarify that 77.8% of patients successfully removed their catheter within a week after LE. 

11. Discussion: Please highlight the pros and cons of LE over conservative management: 

We have added the role of LE compared to conservative management in line 255-260. We hope this helps clarify the pros and cons of LE. 

11. Discussion: For those who are at higher risk for LE failure, what is your suggested strategy?

Refractory lymphatic leakage is difficult to manage. We suggest an early intervention for the high risk patients, as these patients would not respond well to conservative management. Although these patients also are at a higher risk of failure to LE, drainage volume still decreases significantly after LE (as shown in the added figure from question 9). We have further clarified this in out discussion (line 284-286). 

#Reviewer 2: 

1. I would appreciate a more detailed description of the embolization procedure, providing more technical information.

We described the LE process in further detail in line 134-144. Thank you for the suggestion 

2. You only described perioperative variables of the cystectomy. Consider adding other variables, such as the quantity of embolizing fluid injected, etc.

Thank you for your suggestion. However, the quantity of embolizing fluid has not been recorded in our medical record and thus could not be evaluated. Still, we believe that procedure related risks would be minimal as all leaks were visually sealed during embolization.

3. For the multivariate analysis, why did you set 1000 mL as a cutoff? Is there a study that defines that value as a predictor of lymphatic leakage? You should consider using a discrete value, for example, every 100 mL/day, to understand how the probability of the event increases with each increment.

Thank you for the suggestion. We set 1,000mL/day as our cutoff value, as few previous studies illustrate drainage volume of 1,000 – 1,500 mL/day as a cutoff for surgical intervention. As mentioned in the manuscript there is no clearly defined volume for intervention in lymphatic leaks, and we believe that our study could approaching a cutoff value set for surgical management would be the proper starting point. We clarified this in our manuscript (line 177-179).

#Reviewer 3:

1. The authors should describe the PLND template in more detail.

Thank you for the suggestion. We described the PLND template in more detail, which is outlined in line 124-128 of the manuscript.

2. What is the proportion of patients with lymphoceles that underwent LE?

Thank you for pointing this out. As specified in our selection criteria, we intentionally excluded patients who underwent LE due to lymphoceles (line 103-105). First of all, we wished to focus on early leaks, which has few unique characteristics from lymphoceles (outlined in line 69-75). Moreover, most of the patients who presented with symptomatic lymphoceles were treated with sclerotherapy, thus did not undergo LE. 

3. The authors state that (page 6, line 106): “Lymphatic leakage was diagnosed when radiographic evidence of leakage was found in lymphangiography“. Did all patients undergo lymphangiography?

Thank you for your question. Lymphangiography was used during all LE procedures to visualize leaks from the lymphatic vessel. Patients suspected of lymphatic leakage, and those who agreed on LE therefore underwent lymphangiography. When leaks were visualized, we could diagnose post-operative lymphatic leakage, and undergo LE. Patients without leaks shown in lymphangiography, although not included in the study, would not undergo LE. We clarified our manuscript accordingly (Line 111-114).

4. The authors excluded patients with evident lymphoceles on CT before LE. Why were those patients excluded? Why were not all patients included that underwent LE?

As specified in our selection criteria, we intentionally excluded patients who underwent LE due to lymphoceles. The main purpose of the study is to evaluate LE in early post-operative leaks. We further clarified our exclusion criteria to avoid any confusion (line 103-105). Thank you.

5. The size and extent of the lymphoceles are poorly characterized. Please elaborate.

Thank you pointing out. We described the characteristics of the lymphoceles that were found in 5 of our patients in line 225-229. 

6. How was correct catheter placement during drainage of the lymphocele verified?

Thank you for pointing this out. A drainage catheter was placed in the pelvic space before closing the incision, and a post-operative abdominal x-ray was filmed, which confirmed its correct position. We added this description in line 130-132.

7. I suggest revising the statistics of the paper, adhering to the statistical outcome reporting guidelines provided by major urological journals

Thank you for pointing this out. We made adjusted corrections to our table.

8. How were the complications assessed? Was there a standardized approach?

All patients received imaging studies and out-patient follow-ups based on the National comprehensive cancer network bladder cancer guideline (version 4.2024). Image studies and out-patient records within 6 months after LE were reviewed for LE associated complications. We illustrated this in line 160-163.

9. The confidence intervals in the multivariable analysis are incredibly large.

Thank you pointing this out. We do acknowledge this as our limitation, and is mostly due to our small sample size. However, LE is yet an emerging therapy and our sample size is still one of the largest to be published. We believe that future studies with larger sample sizes would provide a clearer result.

---

## [Decision Letter · Decision Letter 1]

29 Aug 2024

PONE-D-24-20978R1Lymphatic embolization for early post-operative lymphatic leakage after radical cystectomy for bladder cancerPLOS ONE

Dear Dr. Heo,

Thank you for submitting the revised version of your manuscript to PLOS ONE. After careful consideration and in light of the reviewers' comments, we feel that your manuscript has merit to get published but requires some minute alterations prior to publication. Therefore, we invite you to submit a revised version of the manuscript that addresses these points raised.

Kind regards,

Mazyar Zahir, MD 

Academic Editor

PLOS ONE

Journal Requirements:

Additional Editor Comments:

**1. **I believe that, given the small sample size and multiple limitations of the study, the conclusions drawn may overemphasize the benefits of lymphatic embolization (LE). I suggest revising the conclusion section of both the abstract and the main manuscript body ( line 323) to clarify that your findings are only suggestive of a possible effectiveness of LE in controlling post-operative lymphatic leaks after radical cystectomy.

**2. **The very high rate of clinical failure in the robotic surgical group warrants further elaboration. It appears that 3 out of 4 patients (75%) in this group experienced LE failure. This might suggest that the surgical team is less experienced with robot-assisted radical cystectomy (RARC) and typically performs surgeries in an open radical cystectomy (ORC) manner, potentially leading to more lymphatic manipulation and leaks (indicating a learning curve effect).

**3. **Another issue lies with the type of urinary diversion (UD). It was observed that 14 out of 45 patients (31.1%) underwent ureterocutaneostomy (UC), a simple UD associated with a much lower risk of lymphatic injury. However, UC is not commonly utilized in developed Western countries. This represents another limitation of your study, particularly in terms of the generalizability of your findings.

**4. **The definition of clinical success needs to be properly referenced.

**5. **I encourage the respected authors to undertake a thorough linguistic and grammatical revision of the manuscript, ideally in collaboration with a native English-speaking physician. Several sentences do not clearly convey their intended meaning. For example, the sentence in lines 271-272, "Here, we found that a maximal preprocedural drainage volume exceeding 1,000 mL/day was an independent predictor of LE success," implies that a higher preprocedural drainage volume correlates with higher success, whereas the opposite is true. The term "success" should be changed to "failure" here.

Additional lines requiring significant grammatical and linguistic revisions include lines 71-75, 81-86, 103-108, 164-172, 183-185, and 294-300.

**6. **Lines 84-86 and 89-90 discuss similar concepts regarding the prevalence of lymphatic complications. I recommend removing one of these sections to improve the flow of the introduction. The introduction is generally good but needs a comprehensive grammatical and linguistic revision.

Reviewers' comments:

Reviewer's Responses to Questions

**Comments to the Author**

1. If the authors have adequately addressed your comments raised in a previous round of review and you feel that this manuscript is now acceptable for publication, you may indicate that here to bypass the “Comments to the Author” section, enter your conflict of interest statement in the “Confidential to Editor” section, and submit your "Accept" recommendation.

Reviewer #1: All comments have been addressed

Reviewer #2: All comments have been addressed

2. Is the manuscript technically sound, and do the data support the conclusions?

Reviewer #1: Yes

Reviewer #2: Yes

3. Has the statistical analysis been performed appropriately and rigorously? 

Reviewer #1: Yes

Reviewer #2: Yes

4. Have the authors made all data underlying the findings in their manuscript fully available?

Reviewer #1: Yes

Reviewer #2: Yes

5. Is the manuscript presented in an intelligible fashion and written in standard English?

Reviewer #1: Yes

Reviewer #2: Yes

6. Review Comments to the Author

Reviewer #1: I would like to thank the authors for addressing all the comments appropriately. I have no further comments.

Reviewer #2: Dear Authors,

Thank you for submitting your revised manuscript. I appreciate the opportunity to review it after your improvements. I don’t have further comments on this paper.

7. PLOS authors have the option to publish the peer review history of their article (what does this mean?). If published, this will include your full peer review and any attached files.

Reviewer #1: No

Reviewer #2: **Yes: **Bignante Gabriele

---

## [Author Response · Author response to Decision Letter 1]

7 Sep 2024

Response to the editor: 

We would like to thank you for evaluating our manuscript and providing feedback to improve our paper. Here are our responses to your suggestion.

1. I believe that, given the small sample size and multiple limitations of the study, the conclusions drawn may overemphasize the benefits of lymphatic embolization (LE). I suggest revising the conclusion section of both the abstract and the main manuscript body (line 323) to clarify that your findings are only suggestive of a possible effectiveness of LE in controlling post-operative lymphatic leaks after radical cystectomy.

Thank you for your suggestion. We made the following changes in line 56-57 and line 322-323. 

2. The very high rate of clinical failure in the robotic surgical group warrants further elaboration. It appears that 3 out of 4 patients (75%) in this group experienced LE failure. This might suggest that the surgical team is less experienced with robot-assisted radical cystectomy (RARC) and typically performs surgeries in an open radical cystectomy (ORC) manner, potentially leading to more lymphatic manipulation and leaks (indicating a learning curve effect).

We appreciate your response and concern regarding the high rate of clinical failure in RARC. RARC is becoming increasingly common in our practice, and is showing promising results in our institution as well. Unfortunately, this is a single-arm study with only 4 patients who received RARC. Thus, a direct comparison between ORC and RARC is difficult. When enough LE cases are collected, we would be able to attempt a study with a control group (conservative management or sclerotherapy vs LE), which would better elucidate the whether ORC or RARC might take a role in the clinical success of LE. Thank you.

3. Another issue lies with the type of urinary diversion (UD). It was observed that 14 out of 45 patients (31.1%) underwent ureterocutaneostomy (UC), a simple UD associated with a much lower risk of lymphatic injury. However, UC is not commonly utilized in developed Western countries. This represents another limitation of your study, particularly in terms of the generalizability of your findings.

Thank you for pointing out the high incidence of UC in our study. UC is carefully considered in patients with severe bowel conditions (IBD, radiation, bowel adhesions, etc.) in our institution, and avoided whenever possible. As a tertiary institution in Korea, patients with complicated bowel conditions are often referred specifically for the choice of UD in MIBC, therefore resulting in a higher rate of UC compared to ICUD or neobladders.

However, as lymphatic leakage is a complication associated with PLND rather than UD, we believe that the standard PLND template applied in all patients would support the generalizability of our study. 

4. The definition of clinical success needs to be properly referenced.

In this study, we defined clinical success according to timing of drain catheter removal, re-intervention or post-procedural complications. This is unique to our study, and allows our study to truly represent patients who are at a higher risk of undesirable results after LE (further illustrated in line 290-299). We have further elucidated this is our Methods section (line 164-166). Thank you for the feedback.

5. I encourage the respected authors to undertake a thorough linguistic and grammatical revision of the manuscript, ideally in collaboration with a native English-speaking physician. Several sentences do not clearly convey their intended meaning. For example, the sentence in lines 271-272, "Here, we found that a maximal preprocedural drainage volume exceeding 1,000 mL/day was an independent predictor of LE success," implies that a higher preprocedural drainage volume correlates with higher success, whereas the opposite is true. The term "success" should be changed to "failure" here.

Additional lines requiring significant grammatical and linguistic revisions include lines 71-75, 81-86, 103-108, 164-172, 183-185, and 294-300.

Thank you for pointing out this issue. We went through a full revision of the manuscript to improve our linguistic errors accordingly. 

6. Lines 84-86 and 89-90 discuss similar concepts regarding the prevalence of lymphatic complications. I recommend removing one of these sections to improve the flow of the introduction. The introduction is generally good but needs a comprehensive grammatical and linguistic revision.

We made the according change in our introduction (line 85-87), and also went through a thorough linguistic revision. Thank you for your suggestion.

---

## [Editor Report · Decision Letter 2]

9 Sep 2024

Lymphatic embolization for early post-operative lymphatic leakage after radical cystectomy for bladder cancer

PONE-D-24-20978R2

Dear Dr. Heo,

We’re pleased to inform you that your manuscript has been judged scientifically suitable for publication and will be formally accepted for publication once it meets all outstanding technical requirements.

Kind regards,

Mazyar Zahir, MD

Academic Editor

PLOS ONE

---

## [Editor Report · Acceptance letter]

15 Sep 2024

PONE-D-24-20978R2 

PLOS ONE

Dear Dr. Heo, 

I'm pleased to inform you that your manuscript has been deemed suitable for publication in PLOS ONE. Congratulations! Your manuscript is now being handed over to our production team.

Kind regards, 

on behalf of

Dr. Mazyar Zahir 

Academic Editor

PLOS ONE